# Will GLP-1 Analogues and SGLT-2 Inhibitors Become New Game Changers for Diabetic Retinopathy?

**DOI:** 10.3390/jcm11206183

**Published:** 2022-10-20

**Authors:** Katarzyna Wołos-Kłosowicz, Wojciech Matuszewski, Joanna Rutkowska, Katarzyna Krankowska, Elżbieta Bandurska-Stankiewicz

**Affiliations:** Clinic of Endocrinology, Diabetology and Internal Medicine, School of Medicine, Collegium Medicum, University of Warmia and Mazury in Olsztyn, 10-900 Olsztyn, Poland

**Keywords:** diabetic retinopathy, glucagon-like peptide 1 receptor agonists, GLP-1 Ras, sodium-glucose cotransporter 2 inhibitors, SGLT-2 inhibitors

## Abstract

Diabetic retinopathy (DR) is the most frequent microvascular complication of diabetes mellitus (DM), estimated to affect approximately one-third of the diabetic population, and the most common cause of preventable vision loss. The available treatment options focus on the late stages of this complication, while in the early stages there is no dedicated treatment besides optimizing blood pressure, lipid and glycemic control; DR is still lacking effective preventive methods. glucagon-like peptide 1 receptor agonists (GLP-1 Ras) and sodium-glucose cotransporter 2 (SGLT-2) inhibitors have a proven effect in reducing risk factors of DR and numerous experimental and animal studies have strongly established its retinoprotective potential. Both drug groups have the evident potential to become a new therapeutic option for the prevention and treatment of diabetic retinopathy and there is an urgent need for further comprehensive clinical trials to verify whether these findings are translatable to humans.

## 1. Introduction

Diabetes mellitus (DM) and its long-term complications have increasingly become a global burden for both patients and healthcare professionals. It is one of the most common diseases worldwide, which despite constantly improving treatment methods, still greatly affects patients’ quality of life; it constitutes one of the leading causes of disabilities and increased morbidity and mortality rates. Diabetic retinopathy (DR) is the most frequent microvascular complication of DM [1], and the most common cause of preventable vision loss [2,3]. Although it has traditionally been recognized as a microangiopathic complication, it has recently been reclassified as a neurovascular disorder that results from the impairment of the neurovascular retinal unit [4]. DR is estimated to affect approximately one-third of the diabetic population, with vision-threatening proliferative diabetic retinopathy (PDR) and diabetic macular edema (DME) occurring in almost one-tenth of patients with DM [3,5]. Together with the increasing prevalence of DM itself, the incidence of DR is also expected to raise from 415 million in 2015 to 642 million by 2040 [6]. Several risk factors for the development and progression of DR have already been strongly established. They include duration of the disease, presence of hypertension, poor glycemic control, dyslipidemia and microalbuminuria [7,8,9]. Hyperglycemia-induced vascular damage and inflammation have long been considered to play a major role in the pathogenesis of retinal microvasculopathy [10]. However, retinal neurodegeneration has recently been proven to independently contribute to the development of DR [11]. 

The current knowledge regarding risk factors and pathogenesis of DR has established the main directions for therapeutic strategies, although treatment of DR still remains unsatisfying and fails to arrest the clinical progression or reverse the existing retinal damage. The available treatment options—intravitreal anti-vascular endothelial growth factor (anti-VEGF) drugs and intravitreal corticosteroids in DME and laser photocoagulation of the peripheral retina in PDR, only address late stages of this complication [12], while in the early stages there is no dedicated treatment besides improving risk factors—optimizing blood pressure, as well as lipid and glycemic control. Thus, much effort is being dedicated to finding novel therapies that could improve patient outcomes. 

Recently, the introduction of novel antihyperglycemic drugs—sodium-glucose cotransporter 2 (SGLT-2) inhibitors and glucagon-like peptide 1 receptor agonists (GLP-1 RAs), due to highly beneficial effects on cardiovascular and renal outcomes, has rapidly revolutionized treatment approaches and both drug groups have become a new treatment standard in current recommendations of diabetes associations [13]. In numerous studies they have undoubtedly demonstrated multi-organ benefits and pleiotropic effects. Would they also be a valuable and beneficial treatment option to prevent or slow down the progression of DR?

## 2. Aim

The aim of this review is to determine whether GLP-1 RAs and SGLT-2 inhibitors could potentially be a valuable treatment option for the prevention or treatment of diabetic retinopathy.

## 3. Material and Methods

This work is based on the available literature. 

## 4. Results

### 4.1. GLP-1 Receptor Agonists

Glucagon-like peptide-1 (GLP-1) is a gut-derived incretine hormone secreted in response to food ingestion. GLP-1 RAs increase physiological response to glycemic stimulus through regulation of insulin secretion, delayed gastric emptying and reduction of appetite [14]. GLP-1 stimulates glucose-dependent insulin secretion acting directly on β-cells, at the same time promoting their proliferation and survival [14]. It also reduces secretion of glucagon by islet α-cells, thereby reducing the risk of hypoglycemia [15]. However, GLP-1 effects are not limited to glycemia, since GLP-1 receptors are widely distributed in other tissues, including the kidneys, cardiovascular system, central nervous system and gastrointestinal tract [16]. GLP-1 receptors expression in the hypothalamus produces increased satiety and reduced appetite, which together with its intestinal effects—prolonged gastric emptying, contributes to a significant reduction of caloric intake and weight loss [17]. Its clinically relevant effects on weight and waist circumference reduction in the overweight and obese, both diabetic and non-diabetic, has led to the approval of one of its representatives—liraglutide, as a weight management therapy [18]. GLP-1 RAs treatment has been also shown to exert positive effects on lipid profile, reducing total cholesterol, low-density lipoprotein (LDL) cholesterol and triglycerides [19]. Presence of GLP-1 receptors in the endothelium, coronary arteries and cardiac ventricles is responsible for the improvement of endothelial function, increase of cardiac output and contractility, reduction of cardiomyocyte apoptosis and reduction of blood pressure in hypertensive subjects [20]. All of these effects are reflected in the strong clinical evidence of GLP-1 RAs having the potential to reduce cardiovascular events, cardiovascular mortality and all-cause mortality [21,22,23]. In addition, they have well-established renal benefits demonstrated particularly by preventing microalbuminuria and reducing urinary albumin excretion, hence slowing down the progression of nephropathy [24,25]. There is much evidence that GLP-1 RAs have important neuroprotective and anti-inflammatory effects in neuronal structures [26,27]. Moreover, GLP-1 receptors have been identified in the human retina and it was proven that this expression is present only in ganglion cell layers of the healthy human eyes with no such presence in the retina of patients with DR and PDR [28]. Thus, the potential benefits of GLP-1 RAs in the prevention and early treatment of DR have become a subject of study. 

#### 4.1.1. Outcomes of Pre-Clinical Studies

Retinal neuroprotective effects of GLP-1 RAs have been demonstrated in animal studies not only for systemic intake, but also topical administration, independently of blood glucose levels [29]. Numerous animal studies have further shown that GLP-1 activation prevents retinal neurodegeneration through inhibition of neuronal apoptosis, promotion of glial cell activation and protection of the blood-retinal barrier, dysfunction of which is the most important pathophysiological mechanism in early DR [30]. Anti-inflammatory effects on the retina and optic nerve were revealed for lixisenatide in mice, unrelated to its systemic action [31]. GLP-1-mediated anti-oxidative effects, that can be attributed to increased endothelial expression of extracellular superoxide dysmuthase, might also have preventive effects on retinal damage [32]. In both in vitro and in vivo rat models with damaged retinal ganglion cells (RGCs), which are the afferent neurons that transmit visual information to the central nervous system and in the early stages of DR prove to be the most vulnerable, use of a GLP-1 analogue attenuated high-glucose induced damage to the RGCs [33]. The suggested mechanism involved preventing mitophagy through the PINK1/Parkin pathway. The study provided strong experimental evidence supporting retinal neuroprotective potential of GLP1-RAs. Similarly, in a recent in vitro study on human retinal endothelial cells, dulaglutide was confirmed to alleviate oxidative stress induced by high glucose concentrations [34]. This effect was produced by restoring telomerase activity and the expression of endothelial sirtiun-1 (SIRT-1), reduction of which promotes cell aging. Consistent results were obtained for liraglutide in an animal model—it revealed the potential to inhibit oxidative stress and endoplasmic reticulum stress, thus exerting a protective effect on retinal neurodegeneration in diabetes [35]. In consequent studies, topical administration of a GLP-1 RA confirmed its anti-oxidative properties through DNA damage prevention and repair improvement, coupled with promotion of cellular proliferation, thus enhancing neuroproliferation [36,37]. Furthermore, GLP-1 topical administration proved to revert a well-established impairment of the neurovascular unit through reduced VEGF expression and anti-inflammatory action [38]. 

Results of the above-mentioned pre-clinical trials assessing the effects of GLP-1 RAs on the retina are presented in Table 1.

#### 4.1.2. Outcomes of Clinical Trials

In contrast to the auspicious results of pre-clinical studies, we are still lacking clinical trials dedicated to the effects of GLP-1 RAs on DR. Since the introduction of GLP-1 RAs and their increasing use, there have been several reports indicating possible deleterious effects on DR, which initiated the discussion on its safety and decreased the initial enthusiasm. In the retrospective study with exenatide the risk of progression of DR was demonstrated [40], but the effect was transient. In the Trial to Evaluate Cardiovascular and Other Long-term Outcomes with Semaglutide in Subjects with Type 2 Diabetes (SUSTAIN-6) increased risk of retinopathy was reported for semaglutide [41], whereas in the Liraglutide Effect and Action in Diabetes: Evaluation of Cardiovascular Outcome Results (LEADER) trial a statistically nonsignificant higher incidence of retinopathy was demonstrated for liraglutide [42]. Analysis of these findings entails limitations that stem from the fact that results were based on adverse events reports and subsequent precise evaluation was not performed. Causal relationship between DR risk and the use of GLP-1 agonists was not confirmed in further meta-analyses [43,44,45,46]. In an AngioSafe study, designed to determine the safety of GLP-1 RAs in the retina, in both clinical models with liraglutide and experimental models with exenatide, negative effects on retinal angiogenesis and severe DR were not confirmed [47]. Similarly, in a recent study designed to evaluate the postulated negative effect of semaglutide on the retina, topical administration of this agent contributed to reduced glial activation, reduced expression of proinflammatory cytokines, as well as decreased vascular leackage, which independently of its antihyperglycemic action proved a beneficial effect on the retina [39]. A subsequent meta-analysis of randomized controlled trials for semaglutide did not demonstrate increased risk of DR, although long duration of DM and older age of patients was associated with higher risk [48]. Another study comparing incidence of DR in patients treated with GLP-1 RAs and other antidiabetic medications did not reveal an overall increased risk for GLP-1 RAs and a 33% reduction was demonstrated for GLP-1 RAs compared to insulin [49].The potential reasons for these inconclusive results and primary unfavourable reports from cardiovascular outcomes trials might result from the rapid improvement of glycemia caused by GLP-1 RAs treatment initiation, a fact which has long been recognized as an early worsening factor of DR, particularly in patients with higher initial HbA1c levels, its higher reduction and preexisting DR [50,51]. In addition, clinical trials designed to assess cardiovascular outcomes did not meet the requirements for DR assessment in terms of follow-up duration and DR grading, which does not allow precise comparison of the results [52]. This is reflected in a meta-analysis of retinopathy outcomes in cardiovascular studies, which demonstrated that retinopathy risk correlated with HbA1c reduction during treatment [53] and current American Diabetes Association (ADA) “Standards of Medical Care in Diabetes” which recommend to determine retinopathy status before treatment intensification [13].

Results of clinical trials assessing the effects of GLP-1 analogues on DR are summarized in Table 2.

### 4.2. SGLT-2 Inhibitors

Sodium-glucose cotransporter 2 (SGLT-2) inhibitors—antihyperglycemic agents, which increase urinary glucose excretion by suppressing its reabsorption in kidney proximal tubules, have proven not only to significantly reduce glycated hemoglobin (HbA1c) with low risk of hypoglycemia and weight loss promoting potential, but also to improve metabolic parameters such as blood pressure, lipid profile and uric acid levels [54]. In patients with type 2 DM it has been shown to reduce serum triglyceride level and increase HDL-cholesterol level [55]. Its insulin-independent glucose lowering mechanism of action ameliorates glucotoxicity and possibly reduces beta cell workload [56], while fast reduction of glucose levels induces glucagon secretion, thus increasing lipolysis and promoting reduction of liver fat and visceral adiposity [57]. SGLT-2 inhibitors have been also proven to preserve and improve renal function, both through its hemodynamic and metabolic effects. They reduce glomerular capillary hypertension and hyperfiltration, hence lowering albuminuria, mechanical stress on glomerular filtration barrier and tubular reabsorbtion oxygen consumption, which together with decreased renal glucotoxicity, accounts for their renoprotective effects [58]. Their well-established cardioprotective properties, both in diabetic and non-diabetic populations, mechanisms of which are not precisely defined yet, result presumably from their diuretic effects with reduced blood pressure and volume retention, as well as early natriuresis, consequent haematocrit increase, improved vascular function, reduction of inflammation, oxidative stress and pro-inflammatory cytokine production and improved cardiac energetics induced by ketone bodies production [59]. Since hyperinsulinemia and increased insulin resistance promote macrovascular and microvascular complications [60], SGLT-2 inhibitors with their highly beneficial metabolic effects could be hypothesized to provide a novel treatment option also for DR.

#### 4.2.1. Outcomes of Pre-Clinical Studies

Sodium-glucose cotransporters have also been identified in the lens and the retina [61]. Although their function is not fully understood, they might play a role in transportation of nutrients to the retina and maintaining integrity and survival of the neurosensory retina [62]. SGLT-2 receptor expression has also been confirmed in bovine retinal pericytes and SGLT-2 inhibitors reduced glucose-induced pericyte swelling and overexpression of the extracellular matrix [63] which might prove their protective effects on the blood-retinal barrier. In the study on spontaneously diabetic fatty rats, treatment with ipragliflozin decreased oscillatory potential on the electroretinogram and inhibited the progression of cataract formation [64]. In type 2 diabetic mice, long-term tofogliflozin treatment significantly improved impaired retinal neurovascular coupling through the inhibition of retinal glial activation, VEGF protein expression in the retina and improved regulation of retinal blood flow [65]. Similarly, in a study conducted on Akimba mice, an established representative DR model, empagliflozin treatment resulted in reduction of microaneurysms, IRMA, neovascular tufts, vessel tortuosity and vascular leakage, as well as significant reduction of VEGF and albumin expression in the retina and retinal genetic signature alteration [66]. Interestingly, in a study comparing microvascular complications in type 1 and type 2 DM groups in murine models, empagliflozin reduced DR in almost 50% of the type 1 DM model, while no such effects were observed for the type 2 DM model [67].

Low retinal adiponectin concentrations that cause increased vascular permeability and have already been suggested to correlate with the development and progression of DR [68] were found to be prevented by dapagliflozin treatment [69]. SGLT-2 inhibitor-dependent low-grade hyperketonemia, resulting in shift of fuel metabolism from glucose and fat oxidation to more energy efficient and less oxidative stress producing ketone metabolism, has been suggested to account for the cardiorenal beneficial effects [70]. Since human retina is a highly oxygen-consuming unit, such a metabolic switch, together with the anti-inflammatory and anti-oxidative properties of ketones, could potentially exert a beneficial effect also for DR [71]. In a recent study conducted in a diabetic mice model this inhibition of oxidative stress and apoptosis, improvement of tight junction in the retina, reduction of inflammation and production of angiogenic factors was demonstrated for empagliflozin, suggesting its potential for the prevention of DR [72]

Owing to the fact that DR has recently been redefined as a neurovascular complication, one research direction aims at identifying therapeutic strategies with neuroprotective potential. Recent findings have confirmed SGLT-2 inhibitors’ neuroprotective properties in terms of central nervous system pathologies and cognitive impairment [73], mechanisms of which might theoretically coincide in DR. Since hyperactivity of the sympathetic nervous system in patients with type 2 DM activates neural damage of the outer layer of the retina, the reduction of its major neurotransmitter—noradrenaline, proven in animal studies for dapagliflozin in the heart and kidney, might exert a neuroprotective effect in the retina [62]. Additionally, a recent study in in vitro and in vivo mice models has revealed that dapagliflozin reduced production of reactive oxygen species and arachidonic acid and attenuated apoptosis of the retina [74]. Another significant effect was reported for dapagliflozin in rats with fructose-induced DM, where it was shown to prevent cataract development [75].

Results of the above-mentioned pre-clinical trials assessing the effects of SGLT-2 inhibitors on the retina are presented in Table 3.

#### 4.2.2. Outcomes of Clinical Trials

Despite these numerous favourable reports on SGLT-2 inhibitor effects in animal models, human data in diabetic eye disease is still limited. SGLT-2 inhibitors were reported to improve chronic DME in patients refractory to standard treatment [76,77] and in vitrectomized patients [78], although the analyzed groups were very small. The diuretic effects of SGLT-2 inhibitors are the suggested mechanism of these beneficial outcomes. Novel data from a retrospective cohort study in patients with DR indicate that SGLT-2 inhibitors might reduce the incidence of DME [79] and there is an ongoing study on the efficacy and safety of combined therapy with SGLT-2 inhibitor and anti-VEGF agent in patients with DME [80]. Additionally, a retrospective study in type 2 DM patients revealed slower progression of DR in SGLT-2 inhibitor-treated patients compared to a sulphonylurea-treated group, independently of glycemic control [81]. Similarly, in another real-world cohort study SGLT-2 inhibitors showed an 11% reduction of the incidence of DR when compared with dipeptidyl peptidase-4 inhibitors [82]. SGLT2 inhibitors were also compared with GLP-1 RAs in a multi-institutional retrospective cohort study in type 2 DM patients in Taiwan and a lower risk of DME was demonstrated for the SGLT-2 inhibitors arm [83]. A recent meta-analysis of randomized controlled trials designed to determine correlation between SGLT-2 inhibitors and the incidence of DR proved a significant reduction of DR risk in patients treated with SGLT-2 inhibitors with diabetes duration of less than 10 years, which confers its potential value in early treatment initiation [84]. Corresponding results were obtained in a meta-analysis of randomized controlled trials, in which ertugliflozin and empagliflozin reduced the risk of DR in patients with type 2 DM [85].

Results of clinical trials assessing the effects of GLP-1 analogues on DR are summarized in Table 4.

## 5. Conclusions

Not surprisingly, GLP-1 RAs and SGLT-2 inhibitors with their pleiotropic effects have become the focus of research as a potential means of prevention and treatment of DR. Theoretically, since both drug groups have a proven effect in reducing risk factors of DR, improvement of glycemic control, blood pressure and lipid profile is already one of the possible mechanisms to reduce the risk for developing DR and its further progression when introduced early in the treatment. Moreover, since retinopathy and nephropathy share pathophysiological mechanisms [86] and there is a defined correlation between severity of DR and diabetic nephropathy [87,88], renoprotective properties of SGLT-2 inhibitors [58] and GLP-1 RAs [89,90] should be translatable to retinopathy. Numerous experimental and animal studies have provided solid evidence for this retinoprotective potential and gave light to the underlying mechanisms. However, there is still a deficit of clinical studies evaluating these effects in humans. For GLP-1 Ras, most of the human data come from cardiovascular outcome trials, the results of which have not met the criteria to appropriately evaluate the effects on DR. Clinical trials with SGLT-2 inhibitors are still too limited to draw robust conclusions. Therefore, it is highly recommended that information on the presence and degree of DR should be incorporated in the randomization process of patients with DM for any future studies with GLP-1 RAs and SGLT-2 inhibitors, regardless of outcomes the studies are aimed at. Owing to the lack of effective prevention of retinopathy and its anticipated increasing incidence, there is definitely an urgent need to design a large-scale prospective clinical study with DR included as a primary or secondary endpoint.

It is also worth mentioning that in one of the pre-clinical trials with SGLT-2 inhibitors unexpectedly more favorable retinoprotective effects were achieved in type 1 DM group. In this regard, further experimental studies addressed to elucidate this issue would constitute an interesting research direction as it would be important for this group of patients. Relevance of topical GLP-1 RAs administration in clinical settings is also an issue to be further clarified, as it could offer an additional option for a more personalized treatment in a wider population of patients.

Further studies are urgently warranted to evaluate these agents’ potential impact on clinical outcomes of DR as we are in a constant race to find new approaches to prevent DR or arrest its progression. Time is vision.

## Figures and Tables

**Table 1 jcm-11-06183-t001:** Retinoprotective effects of GLP-1 RAs in pre-clinical trials.

Authors	Study Design	Outcomes
Yasuda H, Ohashi A, Nishida S, et al. [32] 2016	Exendin-4 in human retinal microvascular endothelial cells	↑ endothelial expression of extracellular superoxide dysmuthase through epigenetic regulation
Hernández C, Bogdanov P, Corraliza L, et al. [29] 2016	Systemic liraglutide and topical liraglutide, lixisenatide, and exenatide in db/db mice and human retinas	↓ glial activation and neural apoptosis through ↓ extracellular glutamate and ↑ prosurvival signaling pathways, independent of glycemia
Shu X, Zhang Y, Li M, et al. [37] 2019	Topical liraglutide in diabetic mice	↓ hyperphosphorylated tau-triggered retinal neurodegeneration via activation of GLP-1R/Akt/GSK3β signaling
Sampedro J, Bogdanov P, Ramos H, et al. [38] 2019	Topical GLP-1 in db/db mice	Reverted reactive gliosis and albumin extravasation, prevented retinal apoptosis, ↓ VEGF expression, ↓ NF-κB, and pro-inflammatory factors
Chung YW, Lee JH, Lee JY, et al. [31] 2020	Lixisenatide-treated retinas vs. untreated or insulin-treated retinas in type 2 diabetic mice	↓ neuroinflammation independent of glycemia
Ramos H, Bogdanov P, Sampedro J, Huerta J, Simó R, Hernández C. [36] 2020	Topical GLP-1 in db/db mice	↓ DNA/RNA damage and activation of proteins involved in DNA repair in the retina -Babam2 and Ki67—a biomarker of cell proliferation.
Liu J, Wei L, Wang Z, et al. [35] 2020	Liraglutide in diabetic mice	↓ endoplasmic reticulum stress through regulation of Trx-ASK1 complex and activation of Erk signalling pathway
Simó R, Bogdanov P, Ramos H, Huerta J, Simó-Servat O, Hernández C. [39] 2021	Topical semaglutide in db/db mice	Prevented diabetes-induced retinal neurodegeneration, neuroinflammation and vascular leakage
Nian S, Mi Y, Ren K, Wang S, Li M, Yang D. [34] 2022	Dulaglutide in human retinal endothelial cells in vitro	↓ high-glucose-induced oxidative stress by restoring the expression of SIRT-1, eNOS and telomerase activity
Zhou HR, Ma XF, Lin WJ, et al. [33] 2022	Liraglutide in retinal ganglion cells in vitro and in vivo in diabetic rat model	↓ retinal ganglion cell damage and mitochondrial damage in ganglion cells caused by high glucose by preventing mitophagy through PINK/Parkin pathway

**Table 2 jcm-11-06183-t002:** Clinical trials assessing the effects of GLP-1 RAs on DR.

Authors	Study Design	Outcomes
Douros A, Filion KB, Yin H, et al. [49] 2018	Cohort study of 77,115 patients with type 2 DM GLP-1RAs vs. two or more oral antidiabetic medication	No association between GLP-1 RAs with incident DR
Gaborit B, Julla JB, Besbes S, et al. [47] 2020	AngioSafe 1: Cross-sectional study of 3348 type 2 DM patients GLP-1RAs and other anti-hyperglycemic medication AngioSafe 2: Randomized open label trial in type 2 DM patients Liraglutide vs. no additional treatment on top of metformin and/or insulin secretagogues	No association with GLP-1RAs and severe DR No effect on circulating hematopoietic progenitor cells—angiogenic markers and angio-miRNAs
Wang F, Mao Y, Wang H, Liu Y, Huang P. [48] 2022	Meta-analysis of 23 randomized controlled trials with semaglutide, 22,096 patients	No increased risk of DR overall, ↑ risk of DR in patients aged ≥ 60 and with diabetes duration ≥ 10 years

**Table 3 jcm-11-06183-t003:** Retinoprotective effects of SGLT-2 inhibitors in pre-clinical trials.

Authors	Study Design	Outcomes
Takakura S, Toyoshi T, Hayashizaki Y, Takasu T. [64] 2016	Ipragliflozin in spontaneously diabetic Torii rats	↓ progression of cataract formation, prevented prolongation of oscillatory potential peaks in electroretinogram
Chen YY, Wu TT, Ho CY, et al. [75] 2019	Dapagliflozin in fructose-induced DM	↓ RAGE-induced NADPH oxidase expression in lens epithelial cells through inactivation of glucose transporters and reduction of reactive oxygen species
Eid SA, O’Brien PD, Hinder LM, et al. [67] 2020	Empagliflozin in in STZ-induced type 1 diabetic mice and db/db mice	↓ retinal degeneration in type 1 diabetic mice, ↓ systemic oxidative stress
Hanaguri J, Yokota H, Kushiyama A, et al. [65] 2022	Tofogliflozin in db/db mice	Prevented activation of glial fibrillary acidic protein and VEGF protein expression in the retina
Matthews J, Herat L, Rooney J, Rakoczy E, Schlaich M, Matthews VB. [66] 2022	Empagliflozin in Akimba mice	↓ vascular leakage demonstrated by ↓ albumin staining in the vitreous humor, ↓ expression of VEGF in the retina
Sakaue TA, Fujishima Y, Fukushima Y, et al. [69] 2022	Dapagliflozin in STZ-induced diabetic mice	Prevented reduction of retinal adiponectin and ↑ vascular permeability
Gong Q, Zhang R, Wei F, et al. [72] 2022	Empagliflozin in db/db mice	↓ branched-chain amino acid accumulation in the retina, ↓ inflammation and angiogenic factors (TNF-α, IL-6, VCAM-1 and VEGF)
Hu Y, Xu Q, Li H, et al. [74] 2022	Dapagliflozin in STZ-induced diabetic mice retinas and human retinal microvascular endothelial cells	↓ apoptosis of the retina independent of glycemia, ↓ production of arachidonic acid in human retinal microvascular endothelial cells, ↓ hyperglycemia-induced apoptosis of human retinal microvascular endothelial cells through inhibition of ERK/1/2/cPLA2/AA/ROS

**Table 4 jcm-11-06183-t004:** Clinical trials assessing the effects of SGLT-2 inhibitors on DR.

Authors	Study Design	Outcomes
Yoshizumi H, Ejima T, Nagao T, Wakisaka M. [77] 2018	Case report 1 patient with steroid-resistant DME	DME recovered
Mieno H, Yoneda K, Yamazaki M, Sakai R, Sotozono C, Fukui M. [78] 2018	5 patients with chronic DME after vitrectomy	↓ central retinal thickness, no change of visual acuity
Cho EH, Park SJ, Han S, Song JH, Lee K, Chung YR. [81] 2018	Retrospective analysis of patient records 21 patients treated with SGLT-2 vs. 71 patients treated with sulphonylurea	↓ risk of DR progression, independent of glycemic control
Chung YR, Ha KH, Lee K, Kim DJ. [82] 2019	Real-world cohort study 41,430 patients SGLT-2 vs. DPP-4	↓ risk of DR incidence, no differences in the risk of DR progression
Takatsuna Y, Ishibashi R, Tatsumi T, et al. [76] 2020	Case report 3 patients with chronic DME resistant to standard treatment	DME improved
Su YC, Shao SC, Lai ECC, et al. [83] 2021	Multi-institutional cohort study 9986 users of SGLT-2 vs. 1067 users of GLP-1	↓ risk of DME
Tatsumi T, Oshitari T, Takatsuna Y, et al. [79] 2022	19 patients with treatment-naive DME	SGLT-2 treatment ↓ central retinal thickness
Ma Y, Lin C, Cai X, et al. [84] 2022	Meta-analysis of randomized controlled trials	↓ risk of DR in patients with diabetes duration < 10 years
Zhou B, Shi Y, Fu R, et al. [85] 2022	Meta-analysis of randomized controlled trials	No correlation between overall SGLT-2i and cataract, glaucoma, retinal disease and vitreous disease Ertugliflozin and empagliflozin ↓ the risk of retinal diseases, canagliflozin might ↑ the risk for vitreous diseases

## Data Availability

Publicly available datasets were analyzed in this study.

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
