# Peer review of "Will GLP-1 Analogues and SGLT-2 Inhibitors Become New Game Changers for Diabetic Retinopathy?"

_jcm, 2022, doi:10.3390/jcm11206183_

Round 1

Reviewer 1 Report

The authors review in detail the current knowledge about GLP-1 and SGLT-2 compounds as potential treatments for DR. While the authors had made an extensie efford to compile the information, the current state of the review (the writting) is shadowing the good work behind it. I would strongly encourage to make significant changes in the writting to improve the message. Some of them are:

I am not familiar with the jounral sections requirements. Is Material and Methods necessary? It confuses me in terms of structure that the information is under M&M.

Citations are not consistent, being most of them in superscript but quite some of them are mixed with normal commas or numbers or just normal numbers.

I strongly recommend significant english improvements and send the manuscript to a native. Some sentences from the very beginning of the manuscript are not easy to understand and I could have done it just because were the very typical statements in the introduction. Some sentences makes no sense or they end abruptly (like line 63). The uses of punctuation are simply incorrect, making the understanding of a sentence very difficult.

I would suggest some spacing between the general effects of GLP-1/SGLT-2 compounds and their effects on diabetic retinopathy, as this is the main part of the review and its difficult to locate, and also very condense-difficult to follow information.

I would tone down some statements such as "disturbing reports". Some sentences are not weritten in very academical english, so please, improve also this.

Author Response

We appreciate the time and effort that you dedicated to providing feedback on our manuscript and are grateful for the insightful comments to our paper. We have incorporated most of the suggestions of the reviewers and made according improvements in the current version.

We have fixed the inconsistent citations which were an oversight. We have re-structured the paper to make it more clear and added tables to summarize the available data according to reviewers suggestions. We have also rewrote the conclusion section and corrected writing and grammar mistakes. 

We hope the manuscript after careful revisions meets your high standards. The authors welcome further constructive comments if any.

Reviewer 2 Report

“The review paper of Wołos-Kłosowicz and colleagues summarised available evidence regarding the use of GLP-1 analogues and SGLT-2 inhibitors in the treatment of diabetic retinopathy. They report both medications have the potential to become a new therapeutic option for the prevention and treatment of diabetic eye disease. The study is well designed. The reviewer has no major concerns with the study design, however has major concerns with the written report. The current study makes a contribution to the evaluation of available medications in diabetic, however not adding any new information or critical discussion to our available established knowledge in this field.  I have major reservations and minor ones which are brought to the attention of the authors in a most constructive manner for their considerations as noted below.”

“First of all, conducting of any review paper has its own specific aim in scientific publication, as these articles are coming to narrow down and simplified the findings of different publications on a detailed goal into a practical message. I feel more confused by reading each section and particularly the conclusion rather than getting any clinical/practical take-home message. Please consider this in your paper to have your unique critical message rather than a very typical ending. Writing a review paper is not just summarising/listing different outcome without providing any critical view. The main goal of present paper is still interesting to me”.

“I would suggest to re-structure this paper by separating pre-clinical (cell and animal studies) and clinical studies with separate headings, for each medication. For example:

1.GLP-1 receptor agonists

1.1 Outcomes of pre-clinical studies

1.2 Outcomes clinical studies

with also providing your own discussion (what we already know and what we need to know and how they can possibly be translated to the clinical use) at the end of each section”.

“I am also surprised that there is no table or figure in this paper (at least in my available review documents). It is necessary to have structured tables to summarised all the mentioned data which are particular of interest for the readers. Reading a review paper without any obvious, structured message and no tables is, honestly, tedious and impractical for most of the readers”.

“Just as an example of rushing to conclusion in this paper is in line:221, they authors listed that human data in diabetic eye disease are EXTREMELY limited and then in line: 241-242, it has been stated that there is strong well-established evidence. I am wondering how you based your conclusion here?. The whole paper is also needed a grammar check too”.

“I highly suggest to re-write and re-consider your conclusion again as it is so neutral and typical ending. The first paragraph of conclusion (line:246-256) is just a repetition and it sounds more “introduction” rather than a conclusion. Also, regarding the future directions for particularly clinical studies in these two medications, please list, in a very confident way, what we need to know and what issues (in details) need to be addressed in the future. please do not list some general, typical, suggestions”.

Author Response

(The authors gave the same response as above.)

Round 2

Reviewer 1 Report

The authors improved substantially the manuscript and, except for the structure being:

1. Introduction

2. Material and methods

2.1. GLP-1 receptor agonists

2.2. Outcomes of pre-clinical studies

2.3. Outcomes of clinical trials

3. SGLT-2 inhibitors

4. conclusions

The point 2 makes no sense, specially in a review. Please modify that. Other than that, manuscript is ready.

Author Response

Thank you for your further comments. We believe these revisions have resulted in a significantly improved manuscript.

Reviewer 2 Report

Thanks for addressing some of the comments. The current manuscript looks better with an acceptable structure.  I still have a major comment regarding your conclusion, please refer to my previous comments.

Also, some basic editing comments about the tables should be noted.

1. Tables should be cited in the body of the manuscript too. For example, "the Retinal neuroprotective effects of GLP-1 RAs have been demonstrated in animal studies (References for the studies) (Table 1)".

2. Your table must have a proper, informative, and independent title. "Table 1. Effects of GLP-1 RAs in pre-clinical trials" . Effect of GLP-1 on ? what markers? 

3. Headings on top of your table below the title. for example,

Study    Treatment      Outcome

4. Please do not copy-paste everything from the body to the table that looks very redundant to the readers. Please summarise the main outcome on DR, rather than listing all other associated/outcomes.

5. I would suggest inserting an arrow symbol instead of using the word. for example, instead of "reduced" insert " ↓ ". These small, but professional, changes would make your table more practical, informative, and easy to follow for the readers.

Best regards

Author Response

Thank you for your further comments. We believe these revisions have resulted in a significantly improved manuscript.

According to your suggestions we have edited tables – all tables were mentioned in the main body of the paper, titles were corrected, headings were added, arrow symbols were inserted. We did not change the body of the tables though, as the main outcomes were mentioned in the text and tables present detailed information for an individual assessment of the reader.

As for conclusion, we have already addressed your initial review. Conclusion has been rewrote and we have indicated future directions for further studies as you suggested.

We hope that you will find the revised manuscript suitable for publication but are happy to consider further revisions.